# Dying with Cancer and COVID-19, with Special Reference to Lung Cancer: Frailty as a Risk Factor

**DOI:** 10.3390/cancers14236002

**Published:** 2022-12-05

**Authors:** Peter Strang, Torbjörn Schultz

**Affiliations:** 1Department of Oncology-Pathology, Karolinska Institutet, Stockholm’s Sjukhem Foundation (SSH), Mariebergsgatan 22, SE 112 19 Stockholm, Sweden; 2Research and Development Department, Stockholm’s Sjukhem Foundation (SSH), Mariebergsgatan 22, SE 112 19 Stockholm, Sweden

**Keywords:** cancer, lung cancer, COVID-19, palliative care services, mortality, place of death

## Abstract

**Simple Summary:**

Frailty has been strongly associated with deaths from COVID-19 in the general population, especially in the elderly, but it is rarely studied in the context of cancer. In this study of 4312 patients who died with cancer, of whom 282 also had concomitant COVID-19, we found that established risk factors, such as older age, male gender, and being a nursing home resident, were also significant risk factors for patients with cancer. Among the risk factors, frailty, as measured using the Hospital Frailty Risk Score (HFRS), was the strongest one, as well as when controlling for age, sex, comorbidities, and socioeconomic factors. In conclusion, frailty should be addressed in cancer care to a greater extent.

**Abstract:**

Older age and frailty have been associated with COVID-19 deaths, but frailty has seldom been studied in the context of cancer. The aim of this paper was therefore to study frailty (measured using the Hospital Frailty Risk Score) and other risk factors in patients who died with advanced cancer and a concomitant COVID-19 infection, with special reference to lung cancer. Of 4312 patients who died with cancer, 282 had concomitant COVID-19 (within the last 30 days), and these patients were significantly older, more often men, and residents of nursing homes. They often had less access to specialized palliative care, and they died more often in acute hospital settings. Patients with cancer who died with COVID-19 were more often frail (57% vs. 45%, *p =* 0.0002), and frailty was independently associated with COVID-19-related deaths, both in univariable and multivariable regression models, as well as when controlling for age, sex, socioeconomic factors on an area level, and comorbidity (measured using the Charlson Comorbidity Index). In the final multivariable model, where patients with cancer who died in nursing homes were excluded, belonging to the high-risk frailty group (OR 2.07 (1.31–3.27), *p* = 0.002) was the strongest prognostic variable in the model. In a separate analysis of a subgroup of deaths due to lung cancer (*n* = 653, of which 45 deaths occurred with concomitant COVID-19), the above associations were not significant, possibly due to too-few cases. In conclusion, frailty is a strong predictor of cancer deaths and should be addressed in cancer care.

## 1. Introduction

The first official cases of COVID-19 were diagnosed in December 2019, and when the first wave of the pandemic occurred after a rapid dissemination of the virus, there was an urgent need to define risk groups. Early in the course of COVID-19, being elderly and male and having hypertension or other cardiovascular diseases, respiratory system diseases, renal failure, diabetes, and dementia were defined as major risk factors, especially in Chinese studies [1,2,3,4].

During the first wave, patients with active cancer were discussed as being a potentially vulnerable group both due to the malignancy itself and immunosuppressive treatments, which has since been verified [5,6,7,8]. The first cancer studies confirmed that age and comorbidity were risk factors for mortality [9], and recent studies on large groups have corroborated that advanced age, male sex, and comorbidities are associated with poorer outcomes, especially in patients with active cancer and ongoing treatment [10,11]. A recent study revealed that unvaccinated patients with cancer still constitute a vulnerable group, highly susceptible to SARS-CoV-2 infections with poor outcomes [12]. However, despite many studies and statistical associations between patients with both cancer and COVID-19 and poorer outcomes with regard to severe illness, there are some controversies. In a recent Mendelian randomization study, Zengbin Li and coworkers concluded that overall cancer, as well as 12 site-specific cancers, including lung cancer, had no casual association with COVID-19 severity [13]. Still, COVID-19-related cancer deaths are numerous, and the stated risk factors, including advanced age, male sex, and comorbidities, are of great importance in understanding and predicting deaths among patients with cancer. In general populations, especially in the elderly, frailty has increasingly emerged as a risk factor [14]. In a British study comprising 1564 patients hospitalized with COVID-19, of whom 425 died, disease outcome was better predicted by frailty than either age or comorbidity [15]. In cancer studies, however, frailty is seldom included. Instead, tumor stage and performance status are recurring variables.

In cancer treatment, Eastern Cooperative Oncology Group Performance Status (ECOG PS) is used to guide treatment selection, as patients with a poor ECOG PS have reduced benefits from different oncological treatments, as well as an increased risk of side effects. ECOG PS was developed from the Karnofsky Index and presented in 1982 as a six-grade scale (values 0 to 5, where 0 corresponds to fully active and 5 corresponds to dead) to be used to support the prediction of response and toxicities related to cancer treatments [16]. As reviewed by Richard Simcock et al., ECOG PS has been used as a prognostic and predictive tool, e.g., to predict the benefits of chemotherapy use and therapy decisions in lung cancer, and it has also been used as a means to assess prognosis in palliative care settings [17]. However, despite its merits, ECOG PS also has limitations, as it is a unidimensional functional score, and the assessments are subjective and, in most cases, dependent on the physician in charge [17]. Although ECOG PS is often used for subgroup analyses of efficacy, it is rarely used in the evaluation of toxicity [18]. Moreover, although ECOG PS is a useful prognostic tool in chemotherapy studies, it is less useful with regard to modern immunotherapies [19], not least in lung cancer, although patients with an ECOG PS of 2 or more are often excluded from studies, which makes it more difficult to draw final conclusions [20]. 

Moreover, ECOG PS fails to recognize important variables, such as comorbidities, cognitive decline, and frailty. A considerable proportion of patients with cancer are aged over 65 years, which suggests that age and age-associated frailty should be included in treatment decisions, as pointed out by, e.g., Takahashi et al. [21]. Frailty is described as a state of increased vulnerability to the poor resolution of homeostasis following stress [22]. This means that a minor stressor, e.g., ordinary cancer therapies, result in a dramatic change in health state, and, for an individual patient with cancer, this might imply a transition from being independent to dependent [22]. Thus, even limited physical stressors in persons with frailty might result in disability, an increased risk of falls, and even in the development of acute delirium. 

Although frailty is strongly correlated with age, frailty is more than normal aging and involves important structural and functional changes in the brain and a decline in the production of important hormones, such as growth hormones, testosterone, and oestradiol, as well as a decline in the production of dehydroepiandrosterone (DHEA) [22]. Changes are also found in the aging immune system, including a defective, low-grade inflammatory response over prolonged periods, resulting in failure to respond to the stress of acute inflammation [23]. Frailty is also associated with frail skeletal muscle metabolism and muscle loss, i.e., sarcopenia, as the muscle homeostasis of protein loss and new muscle formation is affected by the above-mentioned hormonal and immunological factors. Andrew Clegg and coauthors suggest in a well-received position paper in The Lancet that the number of abnormal systems is more predictive than abnormalities in an isolated system, implying that, when physiological decline reaches an aggregate critical mass, frailty becomes evident [22].

Considering these circumstances, measures of frailty should be included in discussions about cancer and COVID-19. Generally, patients fill in frailty tools, of which the Clinical Frailty Scale (CFS) is widely used [24]. There are, however, several patient-administered tools that capture aspects of frailty, but comparisons show that they differ in their content validity, feasibility, and ability to predict all-cause mortality [25]. An alternative way of measuring frailty is to use multiple ICD-10 codes typical for patients with frailty, which enables the retrospective use of comprehensive register data [26]. Therefore, the rationale behind the development of the Hospital Frailty Risk Score (HFRS) was the use of a large number of ICD-10 codes (109 codes in the final version) that are frequently seen in patients with frailty.

Therefore, the aim of this study was to compare the characteristics of those who died with advanced cancer with and without a concomitant COVID-19 diagnosis within the previous 30 days, with special focus on frailty as measured using the Hospital Frailty Risk Score (HFRS), based on ICD-10 codes. The secondary aims were to compare access to specialized palliative care, to compare the proportion of acute hospital deaths, and to perform a separate analysis of a subgroup of patients with lung cancer, as patients with lung cancer were identified early as being a risk group [27].

## 2. Materials and Methods

The Strengthening the Reporting of Observational Studies in Epidemiology (STROBE) criteria were used to report the methods and results [28]. 

### 2.1. Study Design

This retrospective, observational, registry data study was based on the VAL database of the Stockholm region’s central data warehouse. Data were collected for all patients who died with advanced cancer with/without concomitant COVID-19 in the Stockholm region (Stockholm County) from March 2020 to June 2021. 

### 2.2. Population

Inclusion criteria: all patients aged over 18 years who died with advanced cancer with/without concomitant COVID-19 in the Stockholm region (covering approximately 2.3 million inhabitants) from 1 March 2020 to 30 June 2021 were included. Exclusion criteria: patients with missing values with regard to residential address (*n* = 27), which made it impossible to allocate patients to socioeconomic Mosaic groups (see Section 2.3 Variables below), were excluded. The outer limit of 30 June was chosen, as the vaccine effect is difficult to assess in those who died later. As the deaths were based on register data (date of death and ICD-10 codes) rather than death certificates, the following definitions and delimitations were used: Advanced cancer was mainly defined by cancer as the main ICD-10 diagnosis and a concomitant secondary ICD-10 diagnosis of distant metastases. We also included all malignant brain tumors and hematological malignancies, as these malignancies do not present with distant metastases. Moreover, we included all malignant pancreatic tumors due to their poor prognosis. As the majority of all deaths related to COVID-19 occur within 30 days, this time limit was chosen to dichotomize patients with/without concomitant COVID-19. 

### 2.3. Variables

In order to study deaths with and without a concomitant COVID-19 infection, the following explanatory (independent) variables were used: age, sex, the Hospital Frailty Risk Score (HFRS), comorbidities as defined by the Charlson Comorbidity Index (CCI), dementia (as it has been found to be independently related to COVID-19 deaths in previous studies), socioeconomic status on an area level with the aid of Mosaic, access to specialized palliative care (SPC), and living in nursing homes.

HFRS is a measure of frailty, based on 109 weighted ICD-10 diagnoses instead of the collection of data prospectively through patient- or staff-administered frailty instruments [26]. HFRS was developed based on a development cohort of over 22,000 patients with frailty and validated in a cohort of over 1 million patients, including patients with cancer. According to Reference [26], patients with HFRS values of <5 are judged to be non-frail, patients with values between 5 and 15 are frail with an intermediate risk, whereas people with values above 15 are judged to constitute a high-risk group.

CCI is a method of categorizing comorbidities in patients based on the ICD-10 diagnostic codes contained in administrative data, and it is often used as a proxy for comorbidity burden [29]. The retrospective look-back period was one year from the time of death for each patient. In this study, CCI was calculated in two ways: (a) according to the manual and (b) in a variant where malignant diagnoses were excluded in the calculation of CCI, since all patients included in the study died with a malignancy as the main diagnosis.

Mosaic is a system that divides a county or city into socioeconomic areas [29,30,31]. Stockholm County is divided into 1300 small areas, and each area is classified as Mosaic 1, 2, or 3, where Mosaic Group 1 corresponds to the most affluent areas. The three groups are approximately the same size. Mosaic provides socioeconomic information, and based on this, the Stockholm Regional Council can define and distribute different housing areas to the three different socioeconomic classes. This area-based socioeconomic status (SES) is a broad measure of SES that includes classical variables, such as income and education, as well as additional variables, such as lifestyle, cultural aspects, and living arrangements. Therefore, Mosaic groups may be less specific, but they are broader than traditional SES measures. Overall, cluster analyses of more than 40 iterative variables form the basis of the Mosaic groups. Our initial analyses included all the Mosaic groups. However, the data were finally dichotomized based on the data distribution; Mosaic 1 and 2 (i.e., affluent and middle-class areas) were merged and compared with Mosaic 3, i.e., less affluent areas. 

In the Stockholm region, SPC is offered for those with complex symptoms and greater needs, mainly in the form of advanced palliative home care or hospital palliative care units. Both types of care are staffed 24 h a day 7 days a week with physicians, registered nurses, physiotherapists, occupational therapists, dieticians, assistant nurses, and other medical professionals [32].

### 2.4. Selection Bias and Dropouts

Each clinic and care unit must report to the VAL in most cases, even as a basis for their remuneration, which means that the data are close to complete with few missing values. Each person who has used public healthcare during the years studied is included in the VAL database, which also includes most forms of private care since private care providers have economic agreements with the regional council. As persons residing in nursing homes are provided with basic health care by municipal nurses, they have less contact with the health care provided by the county council and might therefore lack some ICD-10 codes important for the calculation of CCI and HFRS. Therefore, these persons were included in the total material, but separate regression models were also calculated when nursing home residents were excluded.

### 2.5. Study Size

Since all deaths from cancer and concomitant COVID-19 between March 2020 and June 2021 were included, no power calculations were performed. 

### 2.6. Statistical Methods and Missing Data

Proportions were compared using *t*-tests and chi-square tests. For correlations, Pearson’s coefficient of correlation was used. Univariable logistic regression analyses were performed for the relevant variables, which were then entered into multivariable logistic regression models. There were few missing data (mainly the Mosaic classification of 27 patients), which were not substituted. The SAS 9.4/Enterprise guide 8.2 was used for a statistical analysis.

### 2.7. Ethics

The Regional Ethical Review Authority (EPN 2017/1141-31) approved this study.

## 3. Results

Of 4312 cancer deaths in patients with advanced cancer, 282 had a concomitant COVID-19 diagnosis (within 30 days). Those with COVID-19 were older (76.4 vs. 73.6 years, *p* < 0.0001) and were more often men (61% vs. 51%, *p* = 0.03), whereas there were no statistical differences regarding distribution in socioeconomic areas (Mosaic groups 1 + 2 versus Mosaic group 3: 67% vs. 68%, *p* = 0.85) or in CCI score (7.1 vs. 6.9, *p* = 0.09). However, in a separate CCI analysis, where all cancer variables were removed, CCI was higher in those with COVID-19 (1.5 vs. 1.2, *p* = 0.02) (Table 1). Patients with COVID-19 also had a somewhat higher (nonsignificant) proportion of dementia (6% vs. 4%, *p* = 0.18) and more often resided in nursing homes (14% vs. 10%, *p* = 0.02). They were admitted to specialized palliative care services to a lesser extent (54% vs. 74%, *p* < 0.0001), and they died more often in acute hospital settings (32% vs. 15%, *p* < 0.0001) (Table 1).

### 3.1. Frailty—Total Cancer Population

The mean HFRS of all patients was 6.1, and it was higher for men (6.4) than for women (5.8) (*p* = 0.0005, data not shown in tables). Frailty in the form of HFRS was assessed both as a linear function (mean value) and in two dichotomized forms. As those in intermediate and high-risk groups are considered frail, the first dichotomization was between these groups versus the low-risk group (HFRS < 5). The other dichotomization was performed to identify the limited high-risk group (HFRS > 15). Those who died with COVID-19 within 30 days of diagnosis had a higher mean HFRS (7.5 vs. 6.0, *p* < 0.0001) (Table 1). The proportions of patients classified as frail (intermediate and high-risk groups), as well as patients who were classified as being in a high-risk group, were higher in those who died with a concomitant COVID-19 infection (patients with frailty 57% vs. 45%, *p* < 0.0002, and patients with high-risk frailty 13% vs. 8%, *p* = 0.0007) (Table 1).

### 3.2. Dying with COVID-19: Univariable and Multivariable Models 

In the univariable models, older age, male gender, and residing in nursing homes were significantly associated with the likelihood of dying with advanced cancer and a concomitant COVID-19 diagnosis (Table 2). Comorbidity as measured using the CCI, dementia, and socioeconomic factors assessed with Mosaic did not achieve significance. Frailty, however, was a strongly significant variable (Table 2).

In the first multivariable model (*n* = 4312), where all variables were loaded, frailty remained a strongly significant variable (intermediate frailty group: OR = 1.36 (1.04–1.78), *p* = 0.02), high-risk frailty group: OR = 1.71 (1.12–2.61), *p* = 0.01), as well as when controlling for age, sex, CCI, dementia diagnosis, socioeconomic Mosaic groups, and residing in nursing homes. Older age and male gender retained their statistical significances in the model, whereas CCI, dementia diagnosis, socioeconomic Mosaic groups, and residing in nursing homes did not (Table 3).

As patients with cancer residing in nursing homes might differ from those living in their own homes, a separate multivariable model was run, where nursing home residents were excluded (*n* = 3886 remaining in the model). In this model, the high-risk frailty group in particular strengthened its significance (OR = 2.07 (1.31–3.27). *p* = 0.002), but age and male sex remained significant (Table 4).

### 3.3. Frailty in Patients with Lung Cancer and COVID-19

Regarding lung cancer, only 45 of 653 patients died with a COVID-19 diagnosis. In total, 38% of all patients were classified as being frail (intermediate + high-risk HFRS), with an equal distribution between those with and without a COVID-19 diagnosis (38% vs. 38%, *p* = 0.97). Frailty was not statistically associated with the limited group of COVID-19 deaths. Numerically, those dying with advanced lung cancer and concomitant COVID-19 were less likely to receive specialized palliative care, but the difference was not statistically significant (76% vs. 84%, *p* = 0.13). Regarding the place of death, 20% and 12% of patients with and without COVID-19, respectively, died in acute hospital settings (*p* = 0.11) (Table 5).

## 4. Discussion

In agreement with previous studies in general populations, older age, male gender, and being a nursing home resident were variables related to a higher likelihood of dying with advanced cancer and concomitant COVID-19 [1,2,14,33]. However, in contrast to larger studies in general populations [2,33], dementia was not independently associated with COVID-19 deaths, possibly due to the few cases of dementia in our study.

Being a resident in less advantaged socioeconomic areas (Mosaic 3) and comorbidities as measured using CCI were not related to deaths with cancer and COVID-19, which is partly surprising. In previous studies that also included Mosaic as a tool to quantify socioeconomic status, patients from less advantaged socioeconomic areas were at a higher risk of COVID-19 deaths [31,34,35]. This disparity between the general population and the cancer population, especially when using the same methodology (Mosaic) in the same catchment area (Stockholm County), is surprising. In one of our first studies, belonging to Mosaic group 3, was, in fact, strongly associated with COVID-19 deaths in the general population [31]. A possible reason for this was that residents in Mosaic 3 areas more often were employed as, e.g., taxi or bus drivers, having a lot of daily contact with the public, and they lived in more crowed, smaller apartments. However, patients with cancer were identified early as a risk group, and it is possible that similar safety measures for patients with cancer were put into place regardless of socioeconomic living conditions.

The lack of correlation between CCI as a measure of comorbidities and COVID-19 deaths in patients with cancer was more surprising, as CCI has been presented as a risk factor for COVID-19 deaths, e.g., in elderly patients in Sweden [14]. However, cancer and cancer metastases are heavily weighted in CCI calculations, and as all patients in this study suffered from advanced cancer, actual differences may have been blurred. Therefore, we calculated CCI where all cancer variables were excluded, which resulted in a small but statistically significant difference: patients who died with COVID-19 had somewhat higher modified CCI values, whereas frailty as measured using HFRS was strongly related to COVID-19 deaths.

Although comorbidities and frailty are related constructs, they do not overlap [26]; in our own comparison, the Pearson coefficient of correlation between CCI and HFRS was only 0.24. Experts in frailty research, such as Clegg et al., believe that frailty is an independent concept distinct from comorbidity and disability and that it might be present in patients without known comorbidities [22]. 

### 4.1. Frailty as a Risk Factor

In both the univariable and multivariable comparisons, frailty was independently associated with the likelihood of dying with advanced cancer and a concomitant COVID-19 infection. Frailty as measured using HFRS was an independent risk factor in the total material, as well as when including patients with cancer who died in nursing homes. Moreover, the odds ratio (OR) for the high-risk HFRS group was strengthened in the model where nursing home residents were excluded and when controlling for variables such as age, sex, and comorbidities, which is well in line with data from general populations [15]. The OR for HFRS was even marginally higher than that for age in this model. 

Frailty was also separately assessed for patients with lung cancer, as frailty has been associated with increased mortality in this patient group [36]. As stated elsewhere, patients with lung cancer might constitute a frail, vulnerable group for COVID-19 infections, considering that lung cancer is associated with older age, significant cardiovascular and respiratory comorbidities, and smoking-related lung damage [37]. However, due to the limited number of patients in our study, it was not possible to establish statistical associations for this subgroup.

### 4.2. Frailty in the Oncologic Setting

As frailty has been widely discussed in connection with COVID-19 deaths in the elderly in general, it makes sense to also study the impact of frailty in the context of cancer, as in this study. As already mentioned in the Introduction, frailty and other geriatric assessments should be included in routine oncological decision making, especially in patient groups such as lung cancer, where more than one-half of all newly diagnosed patients are over 70 years of age [38]. As reviewed by Schulkes et al., among patients with a seemingly good ECOG PS, the prevalence of geriatric impairment is relatively high with regard to both physical and cognitive domains [38]. In a systematic review of frailty in older patients with cancer, Handforth et al. concluded that frailty is associated with poorer outcomes with regard to treatment complications, as well as mortality [39]. Oncologists and lung cancer specialists regularly see elderly patients with a seemingly good ECOG PS who nonetheless develop severe complications after standard chemotherapy treatment. Such a sudden deterioration in general condition is possibly explained by a good ECOG PS but a concomitant, underlying state of frailty. 

### 4.3. Frailty Based on ICD-10 Codes

In clinical use, patient-reported frailty assessments, e.g., the Clinical Frailty Scale (CFS) or the Fried frailty phenotype, are often recommended [24,25,40]. At present, number of different frailty scales are available, but, although they aim to cover aspects of frailty, these measures only show moderate agreement [25]. Further drawbacks are that patient-reported assessments need to be carried out prospectively, they might be too complicated to use in acute care settings, they require a degree of manual assessment, and they tend to be time consuming and subject to inter-operator error [26]. Thus, they are only used in limited subgroups of patients, which hampers the possibility to perform retrospective analyses on total patient populations. Moreover, different frailty scales only cover a limited number of questions, whereas experts in frailty research, such as Clegg et al., emphasize that the number of abnormal organ systems is more predictive than the decline of a specified organ system [22]. As HFRS is based on 109 ICD-10 codes and thus is applicable in retrospective register studies, it has the potential to be used for explorative studies that intend to map the impact of frailty in cancer settings. 

Generally, older patients with cancer, as well as patients with an ECOG PS of 2 or more, are under-represented in cancer trials; therefore, we do not have a deep understanding of vulnerability in these patient groups. HFRS is an alternative method of studying frailty in total cancer populations regardless of age. Such studies might give a basis for the development of treatment guidelines, which is of special interest today with new, emerging treatment options in the form of immunotherapies. With the aid of HFRS, effects, adverse effects, and survival could be assessed in relation to frailty as an independent variable alongside age and ECOG PS. In the era of the COVID-19 pandemic, frailty is certainly an aid in the calculation of mortality risk in different cancer populations.

### 4.4. Strengths and Limitations

The design with complete registry data with few missing values is a strength of this study. As reporting to the VAL databases is mandatory and a basis for remuneration, the data have very few missing values.

A possible limitation to this study is that the diagnosis of each patient was not based on their death certificate but rather on their primary diagnosis during the last episode of care. However, the primary diagnosis was strengthened by a diagnosis of secondary tumors (metastases). With regard to COVID-19 diagnoses, we only included diagnoses made within 30 days of death, as the absolute majority of COVID-19-related deaths occur within this time frame, although deaths attributed to COVID-19 infections after 30 days exist. Moreover, we limited the study period to March 2020–June 2021 in order to cover the first, second, and third waves of COVID-19. However, we refrained from including patients after June 2021, as most patients with cancer were vaccinated during the first half of 2021. As we did not have access to death certificates, we could not judge whether a newly diagnosed COVID-19 infection was relevant to the actual death in vaccinated persons.

## 5. Conclusions

This study verified that the general prognostic factors for dying of COVID-19, such as older age, male gender, and being a nursing home resident, were also prognostic in the context of cancer. Frailty, a strong prognostic factor in general populations, also proved to be the strongest factor with regard to cancer deaths. These findings could, however, not be verified in a subgroup analysis of lung cancer due to the limited number of cases.

In conclusion, frailty had a greater predictive value than comorbidities, and it should be addressed in cancer care, in addition to ECOG PS, when planning treatment, as frailty might be a way to identify persons with an increased risk of adverse effects and mortality.

## Figures and Tables

**Table 1 cancers-14-06002-t001:** Characteristics of patients aged >18 years who died with advanced cancer ^a^ (*n* = 4312) in Stockholm County from 1 March 2020 to 30 June 2021.

Variable	Total *n* = 4312	COVID-19 within 30 Days *n* = 282	Others *n* = 4030	*p*-Value ^b^
Age, all, years, mean (SD)	73.8 (12.1)	76.4 (11.8)	73.6 (12.1)	0.0001
Women		76.1(13.3)	73.3 (12.6)	0.03
Men		76.6 (10.7)	73.9 (117)	0.002
Sex: Men (%)	2240 (51)	171 (61)	2069 (51)	0.02
Mosaic group 1 + 2 (%)	2912 (68)	189 (67)	2723 (68)	0.85
CCI (sd) ^c^	6.9 (1.9)	7.1 (1.9)	6.9 (1.9)	0.09
CCI (sd), cancer excluded ^d^	1.3 (1.5)	1.5 (1.5)	1.2 (1.5)	0.02
Frailty score ^e^ linear (sd)	6.1 (5.8)	7.5 (6.3)	6.0 (5.7)	0.0001
Intermed. + high risk, *n* (%)	1987 (46)	160 (57)	1827 (45)	<0.0002
High risk, *n* (%)	351 (8)	36 (13)	315 (8)	0.003
Dementia (%)	192 (6)	17 (6)	175 (4)	0.18
Nursing homes (%)	426 (10)	39 (14)	387 (10)	0.02
Admitted to palliative care	3131 (73)	153 (54)	2978 (74)	<0.0001
Deaths in acute hospitals (%)	688 (16)	90 (31)	598 (15)	<0.0001

^a^ Advanced cancer; patients with distant metastases or diagnosis of pancreatic, CNS, or hematological malignancy. ^b^
*p*-values between those who received a COVID-19 diagnosis within 30 days preceding death, versus others. ^c^ CCI = Charlson Comorbidity Index. ^d^ Cancer and metastatic cancer were heavily weighted in CCI calculations. In this calculation, all cancer diagnoses were excluded. ^e^ Hospital Frailty Risk Score (HFRS).

**Table 2 cancers-14-06002-t002:** Odds ratios (ORs) (univariable analysis) for the probability of dying with COVID-19 within 30 days of diagnosis. N = 4312 cases of advanced cancer ^a^, of which 282 died with COVID-19 infection.

Variable	Odds Ratio OR (95% CI)	*p*-Value
Frailty groups ^b^ Low (<5) Intermediate (5–15) High (>15)	Ref. 1.48 (1.14–1.92) 2.06 (1.40–3.05)	0.003 0.0003
Age groups 18–69 years 70–84 years ≥85 years	Ref. 1.42 (1.04–1.92) 2.09 (1.47–2.97)	0.003 0.0003
Sex Female Male	Ref. 1.46 (1.14–1.87)	0.003
Mosaic socioeconomic groups 1 + 2 3	Ref. 1.02 (0.80–1.33)	0.85 (ns.)
CCI (linear) ^c^	1.056 (0.991–1.124)	0.09 (ns.)
Dementia	1.41 (0.85–2.36)	0.19 (ns)
Nursing homes No Yes	Ref. 1.51 (1.06–2.15)	0.02

^a^ Advanced cancer; patients with distant metastases or diagnosis of pancreatic, CNS, or hematological malignancy. ^b^ Hospital Frailty Risk Score (HFRS). ^c^ CCI = Charlson Comorbidity Index.

**Table 3 cancers-14-06002-t003:** Odds ratios (ORs) (multivariable analysis *model 1*, all patients) for the probability of dying with COVID-19 within 30 days of diagnosis. N = 4312 cases of advanced cancer ^a^, of which 282 died with COVID-19 infection.

Variable	Odds Ratio OR (95% CI)	*p*-Value
Frailty groups ^b^ Low (<5) Intermediate (5–15) High (>15)	Ref. 1.36 (1.04–1.78) 1.71 (1.12–2.61)	0.02 0.01
Age groups 18–69 years 70–84 years ≥85 years	Ref. 1.32 (0.97–1.81) 1.76 (1.20–2.57)	0.07 (ns) 0.004
Sex Female Male	Ref. 1.43 (1.12–1.84)	0.005
Mosaic socioeconomic groups 1 + 2 3	Ref. 1.02 (0.79–1.32)	0.89 (ns.)
CCI (linear) ^c^	1.004 (0.941–1.071)	0.91 (ns.)
Dementia	0.90 (0.52–1.58)	0.72 (ns.)
Nursing homes No Yes	Ref. 1.18 (0.80–1.73)	0.40 (ns.)

^a^ Advanced cancer; patients with distant metastases or diagnosis of pancreatic, CNS, or hematological malignancy. ^b^ Hospital Frailty Risk Score (HFRS). ^c^ CCI = Charlson Comorbidity Index.

**Table 4 cancers-14-06002-t004:** Odds ratios (ORs) (multivariable analysis *model 2*, nursing home residents excluded) for the probability of dying with COVID-19 within 30 days of diagnosis. N = 3886 cases of advanced cancer ^a^.

Variable	Odds Ratio OR (95% CI)	*p*-Value
Frailty groups ^b^ Low (<5) Intermediate (5–15) High (>15)	Ref. 1.35 (1.02–1.79) 2.07 (1.31–3.27)	0.04 0.002
Age groups 18–69 years 70–84 years ≥85 years	Ref. 1.35 (0.98–1.85) 1.66 (1.11–2.51)	0.07 (ns) 0.01
Sex Female Male	Ref. 1.43 (1.09–1.86)	0.009
Mosaic groups 1 + 2 3	Ref. 1.09 (0.83–1.44)	0.54 (ns.)
CCI (linear) ^c^	0.995 (0.928–1.067)	0.89 (ns.)
Dementia	0.97 (0.47–2.03)	0.94 (ns.)

^a^ Advanced cancer; patients with distant metastases or diagnosis of pancreatic, CNS, or hematological malignancy. ^b^ Hospital Frailty Risk Score (HFRS). ^c^ CCI = Charlson Comorbidity Index.

**Table 5 cancers-14-06002-t005:** Frailty, palliative care, and place of death of patients with advanced lung cancer ^a^ (*n* = 653).

Variable	Total *n* = 653	COVID-19 within 30 Days *n* = 45	Others *n* = 608	*p*-Value ^b^
Frail (intermediate + high risk) ^b^ *n* (%)	245 (38)	17 (38)	228 (38)	0.97
Admitted to palliative care *n* (%)	546 (84)	34 (76)	512 (84%)	0.13
Deaths in acute hospitals *n* (%)	82 (13)	9 (20)	73 (12)	0.12

^a^ Advanced lung cancer; patients with distant metastases. ^b^ Hospital Frailty Risk Score (HFRS).

## Data Availability

The datasets generated and analyzed in this study are available upon reasonable request.

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
