# Peer review of "Dying with Cancer and COVID-19, with Special Reference to Lung Cancer: Frailty as a Risk Factor"

_cancers, 2022, doi:10.3390/cancers14236002_

Round 1
Reviewer 1 Report
Congratulations for the interesting study, which confirm that in oncological setting there are prognostic factors associated with deaths from Covid-19. Frailty is assessed as a strong prognostic factor in the overall cancer population, but the same result did not translate into lung cancer population. It does not bring a significant result in lung cancer, which represented a minority population in the study (only 653/4423 patients). In my opinion, such a small population, being one of the tumors with the highest incidence and prevalence, cannot be included in the primary endpoint. For this reason, I would put the focus on lung cancer as a secondary endpoint.
Author Response
Thank you for valuable comments and please find our comments in the attached Word-file.

Reviewer 2 Report
Remarks for Authors
This manuscript was well summarized about the association of frailty and advanced cancer death using registry data. This manuscript needs some revision to improve.
Materials and methods
1: The retrospective observational study should clearly describe “eligibility criteria” and “exclusion criteria”. Please include these criterions in the Population, or please make flow chart.
2: I think that early stage malignant pancreatic tumors should be excluded. It promotes heterogeneity of this research population.
3: I want authors to describe “Hospital Frailty Risk Score” in detail in the Material and Methods, because it may be uncommon for many oncologists and it is the key words of this study.
Results
1: Table 2, Table 3 and table 4 contain same variables of “Frailty, Age group, Sex, Mosaic group, CCI, Dementia, Nursing homes”. What does it mean?
2: I want authors to describe the same variables of Table1-4 into Table 5 in order to compare the data of advanced lung cancer patients with total advanced cancer patients.
Author Response

(The authors gave the same response as above.)

Round 2
Reviewer 2 Report
Thank you for my reviewing of author's revised manuscript.
The manuscript was improved and I think it as acceptable.